# Feel the Music!—Audience Experiences of Audio–Tactile Feedback in a Novel Virtual Reality Volumetric Music Video

**Gareth W. Young** [1,*] **, Néill O'Dwyer** [2] **, Mauricio Flores Vargas** [3] **, Rachel Mc Donnell** [3] **and Aljosa Smolic** [4]

1   TRANSMIXR, School of Computer Science and Statistics, Trinity College Dublin, The University of Dublin College Green, D02 PN40 Dublin, Ireland
2   PIX-ART, School of Creative Arts, Trinity College Dublin, D02 PN40 Dublin, Ireland
3   ADAPT, School of Computer Science and Statistics, Trinity College Dublin, D02 PN40 Dublin, Ireland
4   HSLU, Lucerne University of Applied Sciences and Arts, 6002 Luzern, Switzerland
*   Correspondence: youngga@tcd.ie

**Abstract:** The creation of imaginary worlds has been the focus of philosophical discourse and artistic practice for millennia. Humans have long evolved to use media and imagination to express their inner worlds outwardly via artistic practice. As a fundamental factor of fantasy world-building, the imagination can produce novel objects, virtual sensations, and unique stories related to previously unlived experiences. The expression of the imagination often takes a narrative form that applies some medium to facilitate communication, for example, books, statues, music, or paintings. These virtual realities are expressed and communicated via multiple multimedia immersive technologies, stimulating modern audiences via their combined Aristotelian senses. Incorporating interactive graphic, auditory, and haptic narrative elements in extended reality (XR) permits artists to express their imaginative intentions with visceral accuracy. However, these technologies are constantly in flux, and the precise role of multimodality has yet to be fully explored. Thus, this contribution to Feeling the Future—Haptic Audio explores the potential of novel multimodal technology to communicate artistic expression via an immersive virtual reality (VR) volumetric music video. We compare user experiences of our affordable volumetric video (VV) production to more expensive commercial VR music videos. Our research also inspects audio–tactile interactions in the auditory experience of immersive music videos, where both auditory and haptic channels receive vibrations during the imaginative virtual performance. This multimodal interaction is then analyzed from the audience's perspective to capture the user's experiences and examine the impact of this form of haptic feedback in practice via applied human–computer interaction (HCI) evaluation practices. Our results demonstrate the application of haptics in contemporary music consumption practices, discussing how they affect audience experiences regarding functionality, usability, and the perceived quality of a musical performance.

**Keywords:** volumetric video; virtual reality; music; user experience; audio–tactile feedback

## 1. Introduction

Haptic technology can simulate the experience of touch by applying force and vibration to a user, but how useful is this technology for contemporary artistic practices in the 21st century? For musicians, the concept of musical haptics has long explored the relationship between auditory experiences of sound and music and the somatosensory stimulation and perception of acoustic sound-generating musical interfaces (Papetti and Saitis 2018). In modern digital music, multimodal and 3D interactive platforms allow musicians to engage with digital sound generators, giving artists more power and control over their musical creations. Furthermore, with the resurgence of extended-reality (XR) technology (Evans 2018), including affordable computational ambisonics and the practice of volumography, the next generation of musicians has unique control over audience perspectives in this developing area of creative media research.



Beyond the musician, the concept of a 21st-century musical performance has also changed. The medium is no longer a static proscenium performance; it moves beyond via novel immersive technologies, such as augmented and virtual reality (AR/VR). Digital performances on XR devices can reach new audiences via contemporary multimodal AR/VR head-mounted display (HMD) devices. The inclusion of multimodality in music production paves the foundations for novel interaction paradigms. The audience is stimulated by an immersive installation or recital, with musical haptics becoming a unique and cognitively challenging performance element. As an indirect interaction via musical haptics, the audience's presence may also be affected when watching a virtual performance using haptic technology. Therefore, we posit that in designing haptic musical experiences as artistic practice, we should consider how the audience may see, hear, and feel the immersive performance they are experiencing.

Within VR, digital audio workstations (DAWs) can take multiple forms. With the reinvigoration of XR technology, the current market offers several innovative modes of music creativity in 3D computer-generated imagery (CGI) production environments that can be accessed via VR HMDs. Furthermore, haptics is gaining widespread acceptance as a critical component of XR technology, creating a sense of touch in a previously audiovisual-focused technology. Similarly, as new paradigms for home media consumption evolve, artists can engage with their audiences in new and exciting ways. Thus, the role of emergent 3D capture and display systems as instruments for new audiovisual production techniques is a continuously evolving element of music performance.

For an audience, a live musical performance is experienced momentarily and with others, and this experience is challenging to reproduce via copies of the same version without context. While traditional digital technology can capture the audiovisual element without question, it often fails to capture the feeling or intimacy of a live audience (see Figure 1). Even when the listener is unaware of vibrations, they can influence recognizable features such as presence (Cerdá et al. 2012). In VR, soloistic performance experiences and feelings of presence have been documented research interests for many decades. In this manuscript, we seek to question audience experiences of a VR volumetric music video that applied vibrotactile haptic feedback. Moreover, we explore the impact of this feedback on feelings of presence within the presented immersive experience.

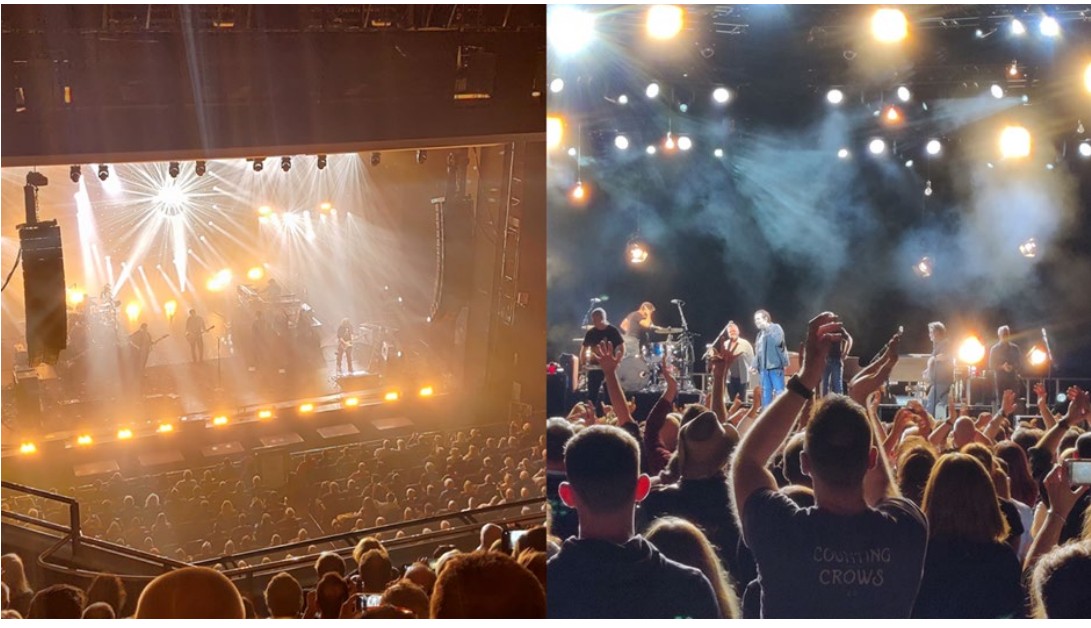

**Figure 1.** Live music performances and audience experiences via the proscenium arch (**left**—The Australian Pink Floyd Show, 2022; **right**—Counting Crows, 2022).

Watching a musical performance is a multifaceted phenomenon intrinsically linked to social, cultural, technical, perceptual, and emotional relationships with music (Gurevich and Fyans 2011). Digital technology has allowed music to transcend physical cause-and-effect paradigms from an audience's perspective, with digital performances becoming CGI interpretations between physical bodies and sound production techniques (Schloss 2003). Today, it is not always possible to bring our audience's understanding of acoustic musical performance to that of a fully mediated digital performance (Gurevich and Fyans 2011). Therefore, perceptibly meaningful connections between action and sound generation are critical for convincing CGI musical performances (Radbourne et al. 2009). Furthermore, the audience's previous knowledge and experience can directly influence their understanding of how a performance platform will work (Fyans et al. 2010).

The "ecosystem" of which a musical performance is delivered comprises four parts: (1) the instrument, (2) the performer, (3) the audience, and (4) the performance environment (Davis 2011). The study of musical haptics has led to a fascination with instrument design and performer-centered studies (Young and Murphy 2015a, 2015b). In traditional HCI evaluations, the role of the audience is often ill-defined. However, comprehensive research has been conducted to capture the more communicative aspects of musical interaction in general DMI practices (Reeves et al. 2005; Fyans et al. 2009; Fyans et al. 2010; Gurevich and Fyans 2011). In the presented works, we define the audience as the listener who watches a performance and has an ancillary relationship with the musical performance process and the performance environment as a 3D CGI virtual reality. In this context, we seek to explore the role of audio-related haptic feedback in audience experiences of a volumetric music video experience presented via VR technology.

Our research aims to explore the relationship between audiences experiencing immersive VR content that applies affordable volumetric video content and interactive elements that seek to immerse the audience in the performance space. To achieve this goal, we created a virtual reality volumetric music video. We explored the audience's perceptions and experiences of such materials, such as their attractiveness, perspicuity, efficiency, dependability, stimulation, and novelty. Furthermore, we display haptic feedback to our participants to discover if this stimulus enhances user experiences and feelings of presence, such as realism, possibility to act, quality of the interface, possibility to examine, self-evaluation of performance, sounds, and haptic feedback. Our hypotheses are as such:

- Haptic feedback, as vibrotactile stimuli, can enhance factors of user experience in virtual reality volumetric music video experiences.
- Haptic feedback, as vibrotactile stimuli, can influence subjective evaluations of the contributing aspects of presence experiences in virtual reality volumetric music video experiences.

## 2. Background and Related Work

Several studies have explored how to augment an audience's experience in live performances, such as theater, dance, and music (Sparacino et al. 1999; Hödl 2016). Researchers have explored new ways for seated audiences to experience embedded actuators in chairs to provide audio-based vibrotactile stimuli (Merchel and Altinsoy 2009; Nanayakkara et al. 2009; Karam et al. 2010). While seating is available in most drama and dance performances, standing is often required for live pop, rock, or dance music concerts. Still, relatively few haptic interfaces are developed for standing-only audiences, with notable exceptions providing free-standing capabilities (Gunther and O'Modhrain 2003; West et al. 2019; Turchet et al. 2021). This factor is significant when considering the contemporary application of immersive technology in musical performance.

VR technology is hardware that harnesses multimodal human–computer interaction to create the feeling of presence in a virtual world (Seth et al. 2011). Thus, contemporary VR employs numerous advanced digital technologies to immerse users in imaginary digital worlds. VR, as technology, is nascent; however, virtual realities, in general, have existed as immersive media entertainment experiences for millennia—as books (Saler 2012; Ryan

1999), films (Visch et al. 2010), theatre (Reaney 1999; Laurel 2013), and games (Jennett et al. 2008). The immersive qualities of such works are often attributed to the quality of the work and not their ability to stimulate multiple senses at once, for example, in the case of vision with film and audio with music. VR experiences are not necessarily modally locked in the same way as other media and can stimulate audiences' senses differently from traditional immersive media.

Haptic cues in music performance and their perception have been observed to affect user experiences—including usability, functionality, and the perceived quality of the musical instruments being used (Young and Murphy 2015b). Haptics can also render and exploit controlled feedback for digital musical instruments (DMIs) (Young and Murphy 2015b). This creative application space highlights the multidisciplinary power of musical haptics from the perspective of computer science, human–computer interaction, engineering, psychology, interaction design, musical performance, and theatre. Therefore, it is hoped that the presented study will contribute to developing a multidisciplinary understanding of musical haptics in 21st-century artistic practices. The role of supplementary senses in immersive media is often undervalued or misrepresented in reductive, single-sensory approaches to lab-based research. In the wild, audiences do not experience a single stimulus while consuming art; they use all their senses to holistically experience the world of live music performance. A notable example would be the severely deaf percussionist, Evelyn Glennie, who has used vibrotactile cues in their musical performance to recognize pitch based on where the vibrations were felt on the body (Glennie 2015).

## 2.1. Immersive Virtual Environments and Presence

Psychologically, virtual realities are presented as 3D immersive virtual environments (IVEs), digitally providing sensory stimuli that encapsulate the user's senses and creating the perception that the IVE is genuine and not synthetic (Blascovich et al. 2002). IVEs have been used for years to convey virtual realities via CAVE and HMD systems (Mestre 2017). Today, VR technology can be used as an erudite psychological platform for cultural heritage (Zerman et al. 2020), theatre performance (O'Dwyer et al. 2022), teaching (Wang et al. 2021), and empathy building (Young et al. 2021).

The most common concepts in discussions about virtual realities are immersion, presence, co-presence, flow, and simulation realism. Immersion is "the degree of involvement with a game" (Brown and Cairns 2004, p. 1298). Immersion is also a deep engagement when people "enter a make-believe world" (Coomans and Timmermans 1997, p. 6). While some research points to experiencing virtual engagement or disassociation from reality in virtual worlds (Brown and Cairns 2004; Coomans and Timmermans 1997; Haywood and Cairns 2006; Jennett et al. 2008), others consider immersion as a substitution for reality by virtuality and becoming part of the virtual experience (Grimshaw 2007; Pine and Gilmore 1999). Immersion also includes a lack of awareness of time and the physical world, feeling present within a virtual world, and a sense of real-world dissociation (Haywood and Cairns 2006; Jennett et al. 2008). While broad, these definitions of immersion are universally applicable to VR technology. Moreover, it should also be noted that measures of immersion target the technology and not the user's experience of the IVE.

Factors of presence, on the other hand, can be classified as subjective experiences (Witmer and Singer 1998). As an aspect of immersion, presence can indicate if a "state of deep involvement with technology" has been achieved (Zhang et al. 2006, p. 2). Therefore, presence can be defined as a "state of consciousness, the (psychological) sense of being in the virtual environment" (Slater and Wilbur 1997, p. 605). Whether directly or indirectly, immersion is required to induce presence. Furthermore, the social aspect of a virtual experience, as co-presence, is also a factor for consideration (Slater and Wilbur 1997) and a state of "flow." Flow describes the feeling of full engagement and enjoyment of an activity (Csikszentmihalyi et al. 2016; Csikszentmihalyi and Larson 2014) and is strongly linked to feeling present and increased task performance in IVEs (Weibel et al. 2008). VR is driven to pursue simulation realism (Bowman and McMahan 2007). The conscious sense of presence

is modeled by presenting bodily actions as possible actions in the IVE and suppressing incompatible sensory input (Schubert et al. 2001). However, a digital representation does not require perfect rendering to be perceived as physically accurate (Witmer and Singer 1998). Furthermore, objective and subjective realism does not always balance when an audience experiences esthetic art practices.

In creative media practices, the connection between presence and visual esthetics is relatively unknown and could be assessed from an immersive arts perspective on realism as an art movement. The relationship between IVEs and esthetics may imply other consequences, as esthetics is associated with pleasure and positive emotions (Reber et al. 2004; Hekkert 2006). Therefore, assessing the feeling of presence in VR experiences as immersive technologies may induce satisfaction and positive affect. As such, presence measures can be effectively applied in user experience studies for evaluating different artistic virtual realities when presented in IVEs without relying on visual realism for immersion.

Using haptics in VR experiences can help increase feelings of perceived presence (Sallnäs 2010), and the effect of haptics on the presence of virtual objects has also been observed (Gall and Latoschik 2018). Moreover, multimodal IVEs, consisting of video, audio, and haptic feedback, have impacted user expectations and satisfaction levels of professional and conventional users (García-Valle et al. 2017). Therefore, evaluating a haptic experience's design can be taken from an audience, performer/composer, instrument designer, and manufacturer perspective (Barbosa et al. 2015). The goal of each stakeholder is different, and their means of assessment vary accordingly. In the presented works, we look to capture audiences' experiences when experiencing musical haptics within an IVE.

### 2.1.1. Multimodal Stimuli

For this experiment, we present an immersive music video experience that implements multimodal stimuli via VR technology—auditory, haptic, and visual. At the heart of any live musical performance are the visual spectacle and the spatial aural experience. In addition, we can also experience supporting haptic stimuli that relate directly to the musical performance as vibrations. Finally, we use the visual senses to correlate the musician's movements with the music performed for the audience (Barbosa et al. 2012). Within VR, we can manipulate the audience to feel present in a virtual world and present imaginative, interactive narratives to immerse the user in a multimodal musical experience.

### 2.1.2. Volumetric Video

Volumetric video (VV) is a media format representing 3D content captured and reconstructed from the real world by cameras and other sensors similarly commonly used in computer graphics (Smolic et al. 2022). VV enables the visualization of such content with full six degrees of freedom (6DoF). Over the last decades, VV has seen interest from researchers in computer vision, computer graphics, multimedia, and related fields, often under other terms such as free viewpoint video (FVV), 3D video, and others. However, the commercial application has been limited to a few special effects and game design cases. Recent years have seen significant interest in VV, including research, industry, and media streaming standardization. On the one hand, this reinvigoration is driven by the maturation of VV content creation technology, which has reached acceptable quality today for various commercial applications. On the other hand, current interest in extended reality (XR) also drives the importance of VV because VV facilitates bringing real people into immersive XR experiences.

Traditionally, VV content creation starts with synchronized multiview video capture in a specifically designed studio. Figure 2 shows an affordable setup used in the V-SENSE lab in Dublin, which only uses 12 conventional cameras. Larger, more complex, and more expensive studios can have up to a hundred cameras and additional depth sensors (Collet et al. 2015). The captured video and other data are typically passed to a dedicated 3D reconstruction process. Classical VV content creation approaches mainly rely on structure-from-motion (SfM)-type approaches or shape-from-silhouette (SfS)-type approaches. While

SfM relies on features and matching and results in a dynamic 3D point cloud in the first place, SfS computes a volume populated by the object of interest in the first place. Both approaches have their advantages and drawbacks. Pagés et al. (2018) presented a system that combines benefits and addresses the creation of affordable capture setups.

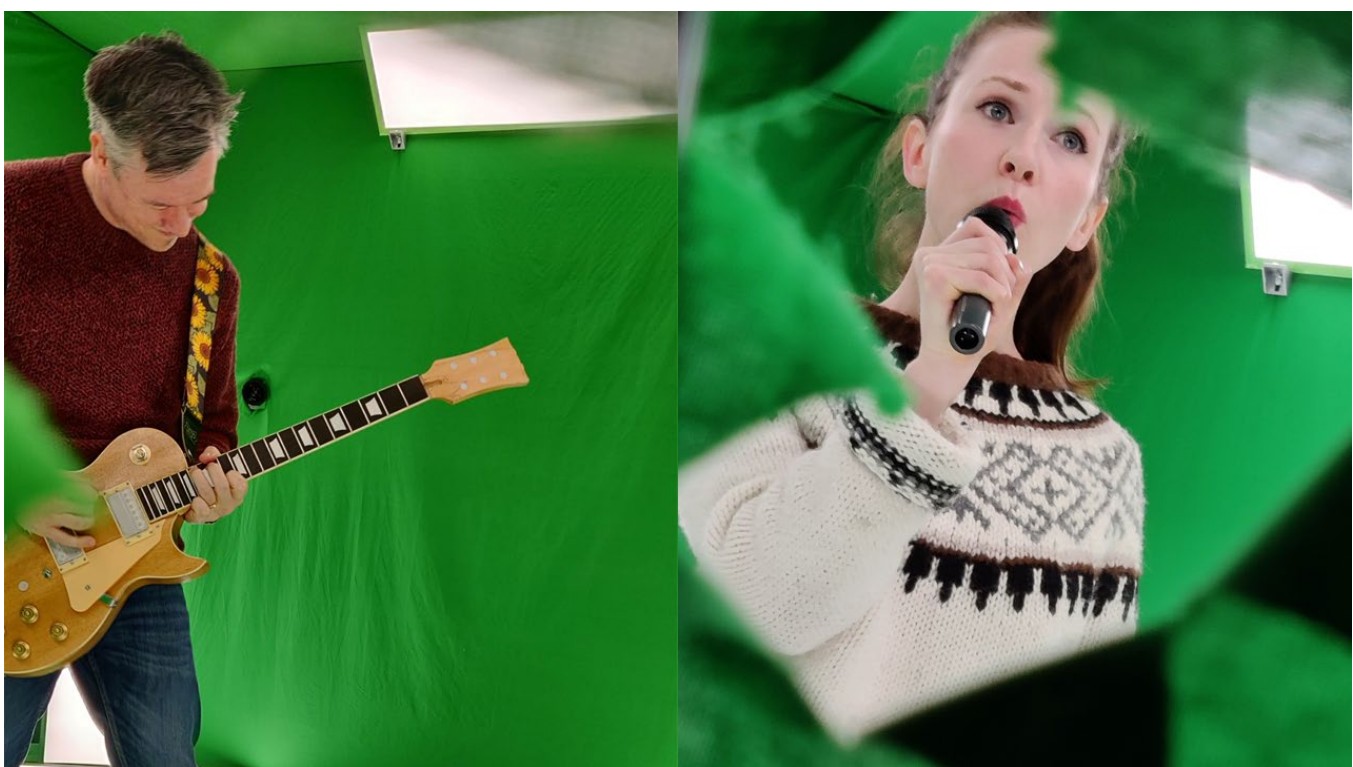

**Figure 2.** Musical performance VV capture by New Pagans[1] Cahir O'Doherty (**Left**) and Lyndsey McDougall (**Right**) at the V-SENSE studio in Trinity College Dublin, Ireland.

Recently, powerful deep learning approaches have been presented for 3D geometry processing and reconstruction (Valenzise et al. 2022). For instance, the first examples of deep learning VV reconstruction algorithms were able to recreate 3D shapes of an object from a particular class of objects, such as a chair, from a single 2D image. A 3D reconstruction of human faces from monocular images or video is another area that has received much attention. PIFu (Habermann et al. 2019) is a single-image 3D reconstruction method of human bodies, representing a milestone in this area. The resulting VV, a dynamic 3D graphics model, can be rendered and visualized for any viewpoint and viewing direction (6DoF), as illustrated in Figure 3. As such, it can be used as an asset in XR content and other media.

### 2.1.3. Spatial Sound

The success of a VR experience relies on effectively replacing real-world sensory feedback with a virtual representation (Slater and Sanchez-Vives 2016). Since sounds convey multiple types of information, such as emotional expression, localization information, and environmental cues, auditory feedback is an essential component in the perception of an IVE. The purpose of auditory feedback in immersive media is to replace the existing sounds and the acoustic response of the environment with virtual ones (Schutze 2018). Furthermore, presence, immersion, and interaction are essential for a successful experience in VR development. The more accurate or plausible the auditory representation, the higher the sense of presence, immersion, and place illusion is felt by users (Avanzini 2022).

Spatial audio, often referred to as immersive audio, is any audio production technique that allows rendering sounds with the necessary perceptual properties to be perceived as

having a distinct direction and distance from the user (Begault 2000; Yang and Chan 2019). Sound localization lets us recognize a sound source's presence, distribution, and interaction (Letowski and Letowski 2012). It is defined as the collection of perceptual characteristics of audio signals that allow the auditory system to determine a sound source's specific distance and angular position using a combination of amplitude, monoaural cues, inter-aural level differences (ILDs), and inter-aural time differences (ITDs) (Bates et al. 2019). Sound auralization is crucial for creating a plausible auditory scene and increasing the user's spatial perception and the VR environment's overall immersiveness. Utilizing a range of acoustic phenomena, such as early reflections and reverberation, allows us to produce a realistic auditory response and helps place audio sources in the virtual space (Geronazzo and Serafin 2022; Yang and Chan 2019).

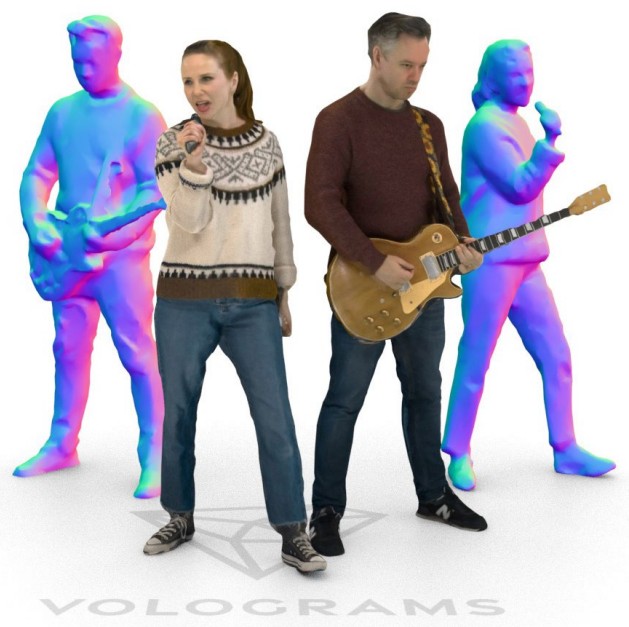

**Figure 3.** A 3D VV example image rendered from different viewpoints of the capture processed by Volgrams[2].

2.1.4. Haptics

The sense of touch in humans is often categorized as cutaneous, kinesthetic and proprioceptive, or haptic perception. Haptic perception is achieved through actively exploring surfaces and objects using the forces experienced during contact with mechanical stimuli, including pressure and vibration. In human physiology and psychology, haptic stimuli and their perception by the brain relate to the actions of the somatosensory system and the sensory gathering of force and tactile information immediately affecting a person, all highlighting the existence of corresponding external stimuli sources. Contact with haptic stimuli is usually made via the skin, explicitly stimulating cutaneous receptors in the dermis, epidermis, and ligament tissue. Cutaneous receptors are found in the skin for touch, and proprioceptors are located in the muscles for kinesthetic and proprioceptive awareness. Cutaneous receptors include mechanoreceptors (pressure or distortion), nociceptors (pain), and thermoreceptors (temperature). Mechanoreceptors need to be stimulated to experience the touch of a vibration.

In physics, vibrations are a mechanical phenomenon whereby oscillations occur around an equilibrium point (Papetti and Saitis 2018). On the one hand, "sound" is a vibration that spreads as an "acoustic wave" via some medium and stimulates the auditory system. On the other, for haptics, the perception of vibration is a measure of vibration as cutaneous stimuli, and this somatosensory information then allows humans to explore their immediate world. For perception to be achieved, direct physical contact is often required;

this is not the case for auditory perception. The radiated sound can also stimulate the surface of the human body. Airborne vibrations, such as sound, can also be perceived by the skin if they are of sufficient amplitude to displace the receptors under the skin, as is often experienced in live concerts.

When an acoustic or digital musical instrument produces a sound, that sound is created by some vibrating element of the instrument's design or an amplified speaker (Figure 4). Therefore, haptics and music can be innately connected through multimodal vibration, where the biological systems of the somatosensory and auditory systems are engaged simultaneously. The combination of haptic and auditory stimuli can be multimodal and experienced by a performer and audience alike, creating new practices that can be mixed and analyzed in multiple contemporary use-case scenarios. The musician and the audience are reached by vibration through the air and solid media, for example, the floor or the seats of a concert space or stage. However, in the case of the audience, vibrotactile and audio stimuli are experienced passively, as no physical contact is made between the instrument and listener. Still, studies have reported that music-related vibrations generally improve the listeners' music experience (McDowell and Furlong 2018; Merchel and Altinsoy 2018), and it is the audience experiences that we choose to observe in this manuscript.

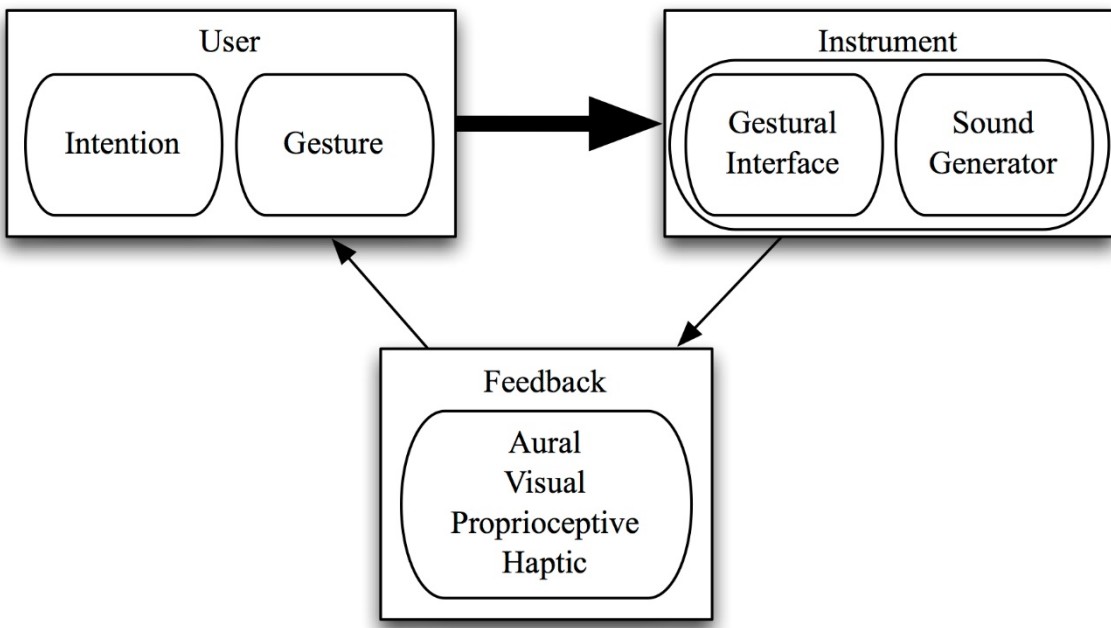

**Figure 4.** The user/instrument/feedback loop of a musical interaction (Young 2016).

2.1.5. VR Performance

The permeation of XR technologies into the hands of creative artists has provoked varied and innovative technological employments toward esthetic ends (Young et al. 2023). The arrival of these technologies has been proposed by several theorists and critics (Bailenson 2018?) as analogous to the advent of film technologies at the beginning of the 20th century, which (arguably) gave rise to the wealthiest epoch of modern, avant-garde, inventive art in the 20th century. Even within the more focused subcategory of the performing arts, there is a plethora of creative techniques, styles, and strategies, as well as opinions and views on the most effective solutions, for harnessing these technologies and captivating audiences. To date, VR (as a subsection of the totality of platforms offered on the spectrum of XR technologies) has enjoyed the most significant level of investigation by performing artists.

Even within the more focused purview of VR performance, several taxonomies still have to be negotiated, for example, live versus prerecorded material and the creative techniques employed. Within the scope of this manuscript, it is suitable to focus the

discussion on VR performance content created using VV, yet even within this narrowed category, there are varying techniques: those that purely use computer vision (V-SENSE 2019; O'Dwyer et al. 2021) and those that include the use of depth camera data (Wise and Neal 2020). Focusing specifically on offline VV content generated purely through the computer vision techniques outlined above, it is essential to note that, in the context of the presented research, there is currently no possibility of generating a live (real-time) representation of a 3D character. Leaving aside consumer bandwidth, the postproduction processes are currently too slow and memory-intensive; however, as processing capabilities increase and algorithms and pipelines become more refined, it is possible that, in the next few years, the latency between capture and representation may be reduced to less than a minute, which is not that far off the latency associated with straightforward video webcasting.

## 3. Methodology

This study aimed to explore an audience's experiences viewing a volumetric music video presented in VR with and without vibrotactile feedback. This process involved observing and evaluating music video audiences individually to gather data on their experiences when engaging with such materials. Thus, A/B testing was implemented to capture this data. The Research Ethics Committee approved the following experiment methodology.

Before conducting our study, we ran an a priori power analysis using GPower to determine an appropriate sample size. We chose an effect size of 0.53, accounting for small and medium effect sizes, to compute a proper range for the sample size, with an alpha error probability $\alpha = 0.05$ and power $\beta = 0.8$. The correlation among measures was left at a default value of 0.5. The power analysis revealed that we would need 20 participants to obtain a medium effect size. We also considered the experiment design recommendations of Macefield (2009), where for comparative studies where statically significant findings are being sought, a group size of 8–25 participants is typically valid.

Recruitment took place in the 2022/23 semester. A general call for participation was made through the project and university networks; additional contributions were also sought from the public via direct email invitations. Volunteers were enrolled across a broad spectrum of potential XR users and were individually invited to experience a VV XR music video on a one-to-one basis. The aims of this research and experiment procedures were shared in advance via a research information sheet, and any questions were addressed. All participants presented with normal or corrected-to-normal vision.

The VR experiment was conducted on a Dell Arora PC with an Intel Core i7 processor and a dedicated NVIDIA graphics card. A Valve Index VR system was used to immerse the participant, with dual $1440 \times 1600$ RGB LCDs running at 120 Hz and a specified field of view (FoV) of 130°. Valve Index wireless controllers were used as the input device. The VV content was developed and run using the Unity3D game engine. Concerning the risk of COVID-19, the HMD was treated with a hydrophobic nanotech coating and hygienically sterilized using a UVC HMD cleaning device (CleanBox X1). Additionally, all touch surfaces and hand-held devices were cleaned with antiseptic wipes before and after each session.

### 3.1. Experiment Task and Measurement Tools

The experiment task was designed with two stimulation factors:

- Baseline Scenario (B0)—participants view a volumetric music video via a VR device;
- Experiment Scenario 1 (S1)—participants view a volumetric music video via a VR device with an additional vibrotactile stimulus.

These two scenarios were used to deliver stimuli for analyzing audience experiences of vibrotactile feedback in VR using a previously validated glove device (see Figure 5) (Young et al. 2013, 2018).

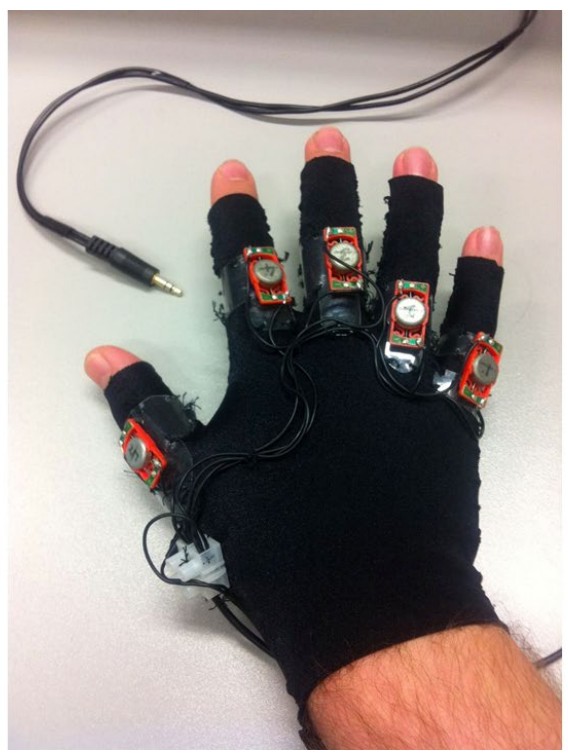 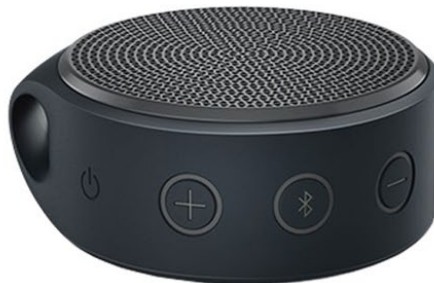

**Figure 5.** Vibrotactile devices (**left**—audio–tactile glove; **right**—Logitech X100).

Haptic technology in musical applications often creates computer-generated virtual experiences by displaying force and tactile stimulus to the user's hands. The use of glove-based devices has been developed for many different musical applications, including passive learning (Huang et al. 2010; Giam et al. 2022), teaching motor skills (Grindlay 2007), increasing learning rates (Fang et al. 2022), and physical therapy (Pal et al. 2021).

The audio–tactile glove used in this experiment has $6 \times$ vibrotactile actuators: 1 on each finger and 1 on the palm. Each actuator can produce a tactile resonant frequency range of 150–300 Hz, with a continuous power handling of 0.5 W with a force factor of 1 Tm. In addition to the glove device, a Logitech X100 was worn around the neck, facing the participants' chest. An X100 has an output bandwidth of 150 Hz to 20 kHz with a peak power output of 3.0 W. The first scenario was chosen as a baseline measure and was used to compare to existing research on VV music video experiences. The participants were asked to view one stimulus randomly and then report on their subjective experiences post-task. Each variable was selected as a representative technology for viewing volumetric music videos in VR.

The research team custom built this experiment's VV VR music video experience, as seen in Figure 6. The study of XR music videos has been used to inform the user-centered design of a custom-made VV VR music video experience featuring the New Pagans' track Lily Yeats. The project's pilot study initially highlighted the specific qualities audiences seek while consuming such materials (Young et al. 2022a). This novel application focuses on the new XR experiences and has been demonstrated and well received within the music technology community (Young et al. 2022b, 2022c, 2022d). It exposes and builds upon existing studies focusing on music and technology in use, specifically how users experience music videos presented via 6DoF XR technologies.

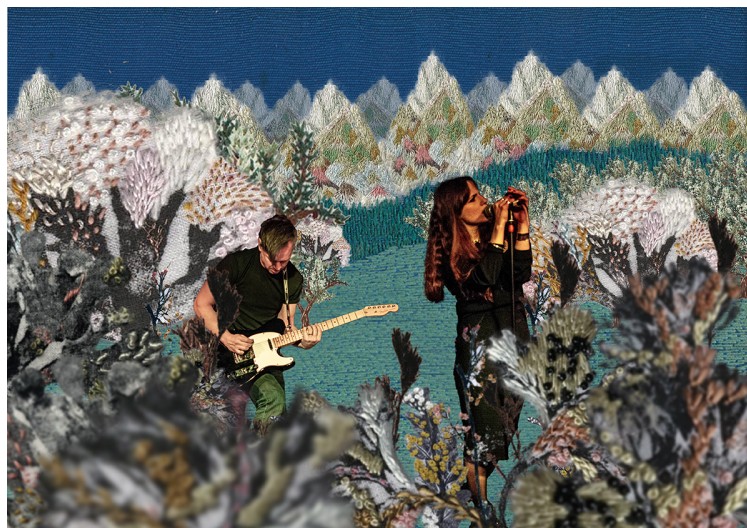

**Figure 6.** Concept scene as part of the design and development stage of the project.

The overall experience takes approximately 5 min from start to finish (https://youtu.be/0Q8zUpefKt8, accessed on 1 January 2023). First, the user is placed within an industrial container yard to orient themselves and become accustomed to the continuous-camera locomotion system. A researcher was present to help the participants become familiar and comfortable with the input controls. Once acclimatized, the user moved through the environment, following audio and visual cues to the first scene transition point. An opening credits scene played at this stage, and the participant was left alone to complete the experience. The main scene required the user to explore the environment and trigger the VV music video experience. This triggering was achieved by collecting a goat's skull from within the stone church and depositing it into the campfire, instructions of which are given "in-game." The VV music experience was then played, and the user could explore the scene and the VV musicians. The scene faded to black at the song's end, and the credits rolled to signify the experience was finished. The researcher returned to the room and helped the participant out of the equipment.

Following the experiment, a post-task questionnaire was delivered via PC. First, the questionnaire captured demographic data, such as age, gender, and musicality, as well as scaling their ability to use digital technology, familiarity with music videos, and expertise using VR on 7-point Likert scales. Then, a User Experience Questionnaire (UEQ) was used to quantify user experiences (Schrepp et al. 2017), capturing the participants' immediate post-task beliefs about using vibrotactile feedback within the context of VR music videos. The questionnaire uses 26 separate question items, each with a seven-stage semantic differential scale, ranging between −3 (extremely bad) and +3 (extremely good). These scales are designed to gather user experience data using different technologies, generating both usability and user experience data. Each scale of the UEQ is mapped to the interaction's overall attractiveness, perspicuity, efficiency, dependability, stimulation, and novelty. The attractiveness rating scales of the UEQ serve as a valence dimension for the goal-directed pragmatic qualities of perspicuity, efficiency, and dependability, and as hedonic qualities of pleasant or unpleasant sensations regarding stimulation and novelty (Laugwitz et al. 2008).

Following this, participants were asked to report their feelings of presence during the experience using a revised Witmer and Singer (1998) Immersive Tendencies Questionnaire (ITQ) using a 7-point Likert scale (UQO Cyberpsychology Lab 2004). The Presence Questionnaire contained 18 items, with attributed subdimensions: realism, possibility to act, quality of the interface, possibility to examine, and self-evaluation of performance. The presence questionnaire included additional references to spatial audio and haptic feedback, giving 23 questions. A high score on the Presence Questionnaire would predict the inten-

sity of a person's sense of presence (Witmer and Singer 1998). Finally, the participant's subjective opinions of the music video content were captured to provide context to the above line of questioning. Furthermore, an open-ended section was provided to gather qualitative feedback and generate a depth of knowledge to support this.

*3.2. Participants*

A total of 20 participants contributed to this study, 9 males and 11 females, with a mean age of 31.68 (SD = 5.61). Nine identified that they were not musical, seven played a musical instrument "sometimes," and four considered themselves musicians. The cohort described their overall digital literacy as "Good" (M = 5.00; SD = 0.84). Their familiarity with music videos (M = 4.20; SD = 0.84) and expertise in VR technology (4.55; SD = 1.02) was used to create a user cube to represent specific user types, as shown in Figure 7 (Nielsen 1994). The distribution of user types was considered representative for further extrapolating our findings.

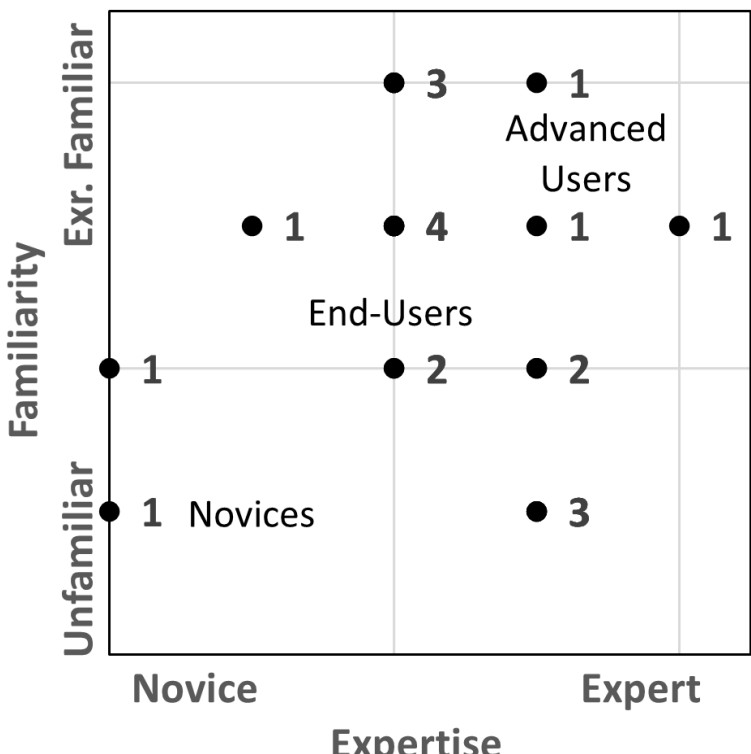

**Figure 7.** User cube showing the distribution of observed user types (inc. number of users).

**4. Results**

Empirical data were collected and analyzed. Qualitative data were coded and used to enrich and add depth of knowledge to our aims and objectives. The analyses of open-ended questions took a thematic approach guided by the frequency and fundamentality of the issues raised by the participants.

*4.1. UEQ*

The UEQ represents the attractiveness and hedonic (stimulation and novelty) and pragmatic (efficiency, perspicuity, and dependability) experiences of each test scenario (see Table 1 and Figure 8).

**Table 1.** UEQ results for B0 and S1 scenarios.

| | **B0** | | | | **S1** | | | | ***T*-Test** | |
|---|---|---|---|---|---|---|---|---|---|---|
| **Scale** | **Mean** | **Std. Dev.** | **Confidence** | **Alpha** | **Mean** | **Std. Dev.** | **Confidence** | **Alpha** | **t** | **Sig.** |
| Attractiveness | 0.55 | 1.03 | 0.64 | 0.92 | 1.57 | 0.80 | 0.50 | 0.85 | 2.46 | 0.02 |
| Perspicuity | 1.08 | 0.67 | 0.41 | 0.49 | 1.65 | 0.84 | 0.52 | 0.80 | 1.67 | 0.11 |
| Efficiency | 0.12 | 0.74 | 0.46 | 0.68 | 0.85 | 0.85 | 0.53 | 0.86 | 0.06 | 0.05 |
| Dependability | 0.33 | 0.78 | 0.48 | 0.61 | 1.00 | 0.86 | 0.53 | 0.71 | 1.84 | 0.08 |
| Stimulation | 0.30 | 1.18 | 0.73 | 0.89 | 1.57 | 0.72 | 0.45 | 0.61 | 2.89 | 0.01 |
| Novelty | 0.73 | 1.09 | 0.68 | 0.90 | 1.69 | 0.54 | 0.34 | 0.64 | 2.51 | 0.02 |

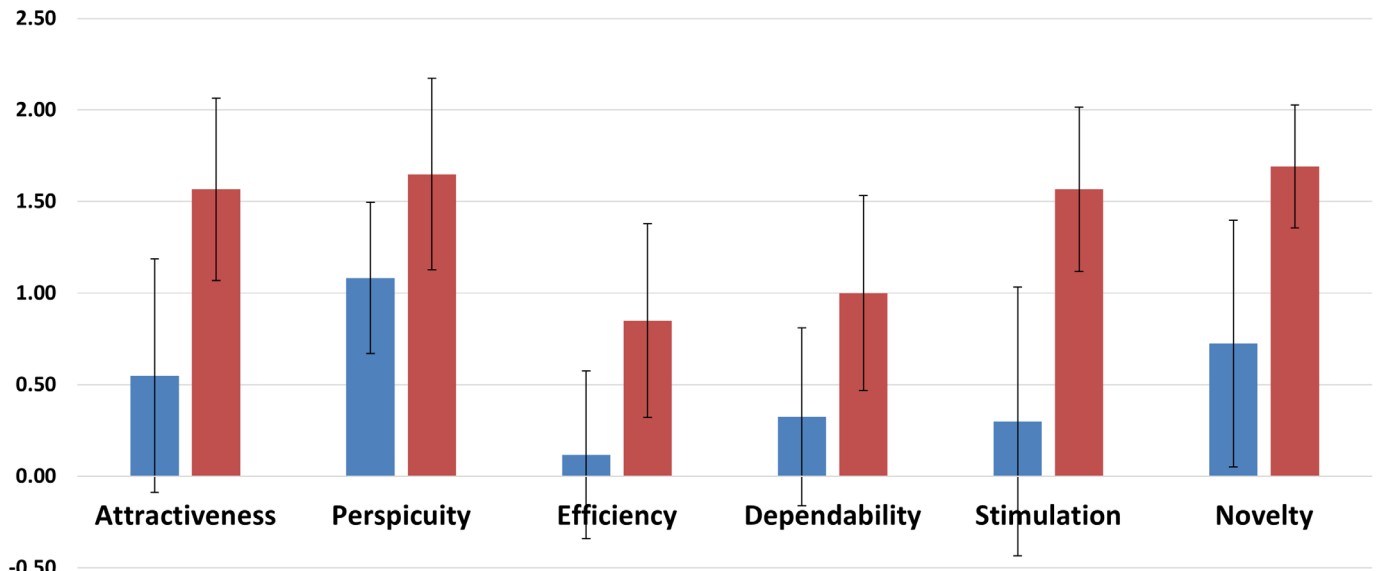

**Figure 8.** Mean UEQ scores for both scenarios (B0—blue; S1—red).

An independent-samples *t*-test was conducted to compare the mean UEQ scores for B0 and S1 (see Table 1). There was no significant difference in scores for B0 and S1 for perspicuity (t (18) = 1.67, *p* = 0.11, two-tailed) and dependability (t (18) = 1.84, *p* = 0.08, two-tailed). The magnitude of the differences in the means for B0 and S1 for perspicuity (mean difference = 1.01, 95% CI: −1.88 to −0.15) and dependability (mean difference = 0.68, 95% CI: −1.45 to 0.96) was moderate (eta squared = 0.07). The *t*-test for evaluating the impact of the intervention showed a statistically significant increase in UEQ scores from B0 to S1 for attractiveness (t (18) = 2.46, *p* = 0.02, two-tailed), efficiency (t (18) = 2.05, *p* = 0.049, two-tailed), stimulation (t (18) = 2.89, *p* = 0.01, two-tailed), and novelty (t (18) = 2.51, *p* = 0.02, two-tailed. Cohen's d statistic indicates a large effect size ranging from 0.76 to 0.98.

*4.2. Presence Questionnaire*

A high score in the presence questionnaire predicts the intensity of a person's sense of presence during an immersive experience. The scale can be separated into subcategories to explore the presence phenomenon further (see Table 2). An independent-samples *t*-test was conducted to compare the total presence scores for B0 and S1. There was no significant difference in total scores for B0 and S1 (t (18) = 0.09, *p* = 0.93, two-tailed). This non-significance was observed for all sub-factors of the questionnaire.

**Table 2.** Mean Presence Questionnaire Scores for B0 and S1.

| | B0 | | S1 | | *T*-Test | |
|---|---|---|---|---|---|---|
| **Presence Questionnaire Categories** | **Mean** | **SD** | **Mean** | **SD** | **t** | **Sig** |
| Total | 114.10 | 14.10 | 113.60 | 11.51 | 0.09 | 0.93 |
| Realism | 29.00 | 7.13 | 26.50 | 5.46 | 0.88 | 0.39 |
| Possibility to act | 18.40 | 3.57 | 19.00 | 3.13 | 0.40 | 0.69 |
| Quality of interface | 16.50 | 2.68 | 17.10 | 3.67 | 0.42 | 0.68 |
| Possibility to examine | 15.20 | 2.35 | 16.80 | 2.57 | 1.45 | 0.16 |
| Self-evaluation of performance | 11.60 | 2.46 | 12.60 | 2.12 | 0.97 | 0.34 |
| Sounds | 16.50 | 2.64 | 16.30 | 3.06 | 0.16 | 0.88 |
| Haptic | 6.90 | 2.42 | 5.30 | 2.79 | 1.37 | 0.19 |

*4.3. Qualitative Feedback*

The qualitative feedback was collected and subject to a thematic analysis using MaxQDA. The inter-coder agreement was kappa = 0.8. A summary of the qualitative feedback can be seen in Table 3. Further cross-analysis of these data is expanded in the Discussion section.

**Table 3.** Qualitative Feedback on the VR Volumetric Music Video.

| Previous Experience | Advantages and Disadvantages | Improvements |
|---|---|---|
| • Nothing or experience with:<br>• Volumetric videos in 2D<br>• Using the Volu app<br>• Cinema<br>• AR/VR HMDs | • New performances<br>• Cost/availability<br>• Analogue vs. digital<br>• Audience perspectives<br>• Audience/artist interaction | • Level of detail<br>• VV shaders<br>• Locomotion<br>• Accessibility<br>• Shared presence |
| **Examples:**<br>"None; I hadn't heard of volumetric music videos." | "If some artists are against hi-tech, they may reject the adaption." | "A greater level of detail and color in the models." |
| "I don't have any experience with 3D music videos." | "People go to a concert for the experience itself, not only for the music and the performance." | "Compared to the live concert, maybe the main difference is the atmosphere." |
| "I have some limited experience using the Volu app." | "Audiences have limited interactions at a live concert." | "A huge group of people's engagement is very different from a person with a headset." |
| "Not sure if Franz Ferdinand's music video counts, but it stuck with me | "I could see it being a barrier for new artists." | "AR/VR concerts offer an inexpensive and accessible way to enjoy performances." |
| "I might only see volumetric scenes from some films." | "I think volumetric music videos introduce costs for artists struggling to break even." | "It feels lonely without the accompany of real people." |
| "Bjork's VR music videos." | "Passivity is sometimes lovely; not everyone will want to have everything interactive." | "The experience was more like an artwork; what it lacks over a real concert is the company of people." |
| "I have seen scans of actors at a conference, and Kpop artist scans for an MR stage performance." | "I feel safer in the front row, but I don't think you can replace a real concert with that." | "I see great potential in music videos because the possibility of interaction is unlimited." |

## 5. Discussion

*5.1. Familiarity with Musical Content*

The cohort was either "Not at all familiar" or only "Slightly familiar" (M = 1.45; SD = 0.8) with the New Pagans' track Lily Yeats. Moreover, they expressed that they "Neither agree nor disagree" (M = 3.4; SD = 0.92) with a preference for this type of music.

This factor reduces the issue of biasing the study with familiarity with New Pagans and preferring this type of music over others.

*5.2. User Experiences*

Perspicuity and dependability were found to have no significant impact on the user's experiences of the VR volumetric music video. This finding means it was relatively easy or difficult for the users to get familiar with the experience, learn how to use the VR equipment, and securely and predictably control the interaction. However, a statistically significant increase in UEQ scores from B0 to S1 was observed for attractiveness, efficiency, stimulation, and novelty. Overall, this outcome means that the general impression of the VR experience was considered more positive with vibrotactile feedback than without it. Pragmatically, the efficiency with which users could solve their tasks without unnecessary effort was more advantageous with the additional input. As the UEQ precisely measures enjoyment, which is loaded on the questionnaire's subscales, we can conclude that, hedonistically, both stimulation and novelty provided an exciting and motivating experience that was fun to use and that the design of the creative experience caught the users' interest.

Vibrotactile feedback has been shown here to impact immersive music experiences, supporting hypothesis one. The mean increase in UEQ scores for attractiveness, efficiency, stimulation, and novelty showed that vibrotactile feedback positively affected participant experiences in both hedonistic and pragmatic areas of the UEQ. However, incidents of perspicuity and efficiency were not impacted significantly. Vibrotactile feedback systems have long been suggested to be beneficial for open-air music controllers based on wearable actuators (Rovan and Hayward 2000). However, more recent research projects have shown a direct positive relationship between musical haptic wearables for audiences (MHWAs) that target audience interaction during live musical performances (Turchet et al. 2021). It is suggested that amplifying vibrations during an immersive music experience can improve the overall musical experience of the audience.

*5.3. Feelings of Presence*

Participants provided insights into the impact of vibrotactile feedback on presence. It was observed that vibrotactile feedback in this context provided no more feelings of presence than without it. To explore hypothesis two further, research is required to address vibrotactile feedback's role and presence measures in immersive musical experiences and position this research within an embodied vs. disembodied interaction using volumetric video and haptic feedback. The subscales of the Presence Questionnaire also identified that only minor differences in the participants' sense of presence could be observed.

It is important to acknowledge various attempts to stage live music and theatre events on social VR platforms, for example, AltspaceVR (AltspaceVR n.d.; O'Dwyer 2021) and VRChat (Scoggin 2021; Benzing 2021) using avatar-based modes of representation and engagement. However, what is gained on social VR platforms concerns co-presence or interaction. Both factors feed into the user's overall sense of immersion and presence. Yet, in the context of "live" versus recorded content, they appear to operate inversely proportional to one another because the bandwidth demands transmitting the real-time (movement and audio) data of several immersants, and sustaining a live conversation (or performance with attendees) erodes the possibility of transmitting expensive data as are necessary for high-fidelity visual representations. Ultimately, the most immersive experiences and realistic simulations will be those that can prioritize both without any noticeable trade-off on one or other side of the scale. Still, more rigorous testing of its psychometric properties and applicability to interactive virtual environments is required (Lessiter et al. 2001).

Concerning presence, the participants directly commented on the spatial haptics and their relationship to audio sources—"It felt amazing to have the music be spatially located; it also appeals further to the user's sense of touch"—"I liked using this technology to experience music, mainly because it was thematic. I could see the musicians and walk

around in an interactive environment." Although the locomotion method chosen provided a more consistent "real-world" motion, providing a "sense of being," some novice users commented on cybersickness. Still, they liked using tech to feel present with art in a novel way and commented that some technical issues could only be solved with more advanced technology and creative use cases. Expressed as such:

> *"VR offers a medium for impressive and awe-inspiring performances; therefore, it shouldn't be wasted on conventional stages (normal-sized people and statues). The live Travis Scott concert in the online video game Fortnite is a good example of what should be aimed for in the virtual scene."*

*5.4. The Use of Volumetric Video*

Many of the participants reported no previous experience with volumetric video. However, some expressed that they had some limited experience. For example, they had used the Volu app, viewed VV on 2D screens, such as cinema or mobile phones, or had directly viewed them via HMDs. Those that had seen VV in practice commented that it was still very experimental, and the quality of the VV was often exemplary of this early stage of development. Furthermore, comments were passed regarding observed cybersickness when displaying VR content in public domains, such as cultural heritage sites. It was suggested that this video capture technology, in 3D and in real-time, would be best suited for AR/VR headsets but could also be improved for home theater experiences. It was also commented that as some of the group had had experience with social VR, a participatory experience would also be exciting. Other comments about one-off experiences with VV include 3D place markers in film production, asset inspection on 2D screens, and building 3D models.

The cohort provided further context when describing the advantages and disadvantages of consuming VV music video content via XR technology involved augmenting the live performance paradigm and enlarging the immediately available audience (see Table 3). Moreover, although expanding, the availability and cost of hardware significantly impede the widespread use of VV to view music performances—"It would be more popular if it were less complex (more streamlined, maybe wireless haptics)." VV allows the audience to walk around, and they may watch the show from their perspective, although it may not be the best-designed perspective. Aside from the perceived quality of volumetric videos, musicians and audiences still have limited interaction in a live concert setting. It could also change how musicians perform as they now need to consider the possibility of the audience viewing their performance from any angle/direction.

It was felt that if some artists are "against hi-tech," they may reject the adaption of "tech Art." Moreover, participants were unsure if the cost was higher to record VV, so it was believed to be a barrier for new artists. For example—"I think volumetric music videos might incur additional charges for artists struggling to break into the music industry." Still, it was believed that simultaneously showing a volumetric video at a concert with a live show could be a good performance idea for an audience. Although, it was highlighted that people go to a concert for the experience itself, not only for the music and the performance. For example:

> *"I think both can co-exist and complete each other, although this new type of performance can benefit musicians (advertising, music discovery, etc.)."*

It was suggested that audiences might want to experience novelty through these videos, so artists might feel compelled to cater to this market—"even though they might not have the necessary technical skill." However, this skill shortage would be a short-term problem, like asking managers to arrange shows and create a digital/social media presence. Regarding directly affecting the live music scene, "it would encourage more people to view the live show since it has higher fidelity and a social aspect that is difficult in VR." However, for some users, accessibility was an important factor—"so live music scenes replicated in VR could be famous for people who cannot attend." As one participant explained:

*"People might prefer concerts in their living room, like Netflix streaming, instead of going to the cinema."*

The participants also commented that volumetric music videos might improve music delivery. The current streaming trend can lead to "music choice overload." However, volumetric music videos might also reduce this fatigue through novelty.

Most comments from participants highlighted the potential to provide access to "live" performances for people who cannot otherwise access them and as a supplement to live performances. The more experienced participants had personally watched virtual artists (like Hatsune Miku) in music videos made for VR. They thought the VR music video concept worked well with the VVs of real artists. However, this could also have its drawbacks—"There's a complete lack of the sense of belonging as there is no audience, low fidelity, and it's uncanny." Moreover, as one participant explained:

*"I don't think VV music videos will affect the live music scene—they cannot yet capture the liveness integral to gigs. It is perhaps more likely that they will compete with commercial music videos and be used similarly."*

The participants suggested that greater detail and color in the models would improve their experiences with volumetric video. This theme highlighted the variations in the capture quality of the VV, as they tended to lose definition in places when the performers were moving fast, a common issue for VV reconstructions. The more advanced users commented that high-definition VV was "expensive" and that the quality of the VV assets was commendable. The fidelity of the VV reproduction was also flagged to the popularity of VV in general. Once VV becomes more representative of the actual performer, it will support the marketplace standing of the medium. However, increasing quality also increases the cost of capturing the performance. For example:

*"They'd need to have a higher definition to become more popular, and they would need to be easily accessible (online hub?) through VR platforms or AR marketplaces on smartphones."*

If the volumetric performances were more personalized and adaptable to the user experience, they would certainly offer an edge over conventional concerts (gaze tracking, reacting to the presence of the user, etc.). To become famous, VV musicians would need to be more accessible, and the experience would have to be more rewarding for everyone. For example, to play music, people can go to YouTube and play a music video. Although VR has been around for a while, engagement is still low. VR and VV can enable experiencing typically inaccessible perspectives—"to experience a metal concert from the mosh pit." It was also noted that what is missing in the VR experience would be a sense of "crowd" or other human factors, though these could also be simulated in VR.

*"You lose part of the immersion, brushing shoulders with another human being at a concert. It's odd watching a performance when you cannot physically interact with the world. However, the vibrations on my hands did improve my immersion overall."*

Compared to the live concert experience, the main difference highlighted by the participants was the atmosphere—"It feels like only myself, and without accompanied by real people." The VV music experience was "good for people who can't go to concerts for many reasons (money, fear of the crowd, access, etc.)." Moreover, the user gains closeness to the artists but loses, somewhat, the modern music video esthetic with fast cuts, etc., they also lose the crowd's feeling in a concert setting. VV music videos can facilitate remote attendance and may be used by those with additional access needs, e.g., wheelchair users, though this may require relatively expensive gear and expertise. The capacity to control your experience of the event is gained through VR volumetric music videos, though the energy from both the crowd and performers is lost. As one participant expressed:

*"A huge group of people's engagement is very different from a person with a headset. A live concert is more touching and exciting, and VR is more supervised."*

However, instead of attending an expensive concert that may require travel, AR/VR concerts offer an inexpensive and accessible way to enjoy performances. It would significantly help people who do not have the time to travel to or sit through entire shows. Volumetric music videos allow the audience to attend and experience the performance and avoid crowds. It will enable more flexibility and more viewing options for the users, but at the cost of the other environmental aspects and the atmosphere of a regular concert, such as crowds (based on user preferences). Similarly, participants highlighted that VV would be "great for nostalgia," i.e., experiencing shows that occurred in the past.

> *"I can imagine there would be a market for people to attend a Beatles concert or Elvis Presley concert since these are things we can't do anymore, but people can participate in regular shows or watch them on TV."*

## 6. Constraints and Future Work

The primary constraint of this study was recruiting a more significant number of participants. Unfortunately, recruitment took longer than pre-COVID-19 due to apprehensions about using HMDs. However, once participants were assured that using UVC LED devices for sterilizing our equipment was safe, they were much more comfortable recommending the experiment to others. More participants would be required to increase the effectiveness of our analysis and improve the statistical significance of this research.

The significance of haptics in the feeling of presence revealed some exciting findings but was ultimately elusive in defining which of the many presence factors was compelling for our participants. Therefore, further presence-focused user studies must be undertaken to explore the role of musical haptics in VR VV music experiences.

A shared virtual environment will be explored in future iterations of this analysis. Moreover, more interactive audiovisual elements will be incorporated to entertain the user further. Our participants suggested these two factors to improve the overall experience. Given the technicality of designing and creating these types of interactions, additional development time is needed to implement more advanced user interactions and social VR systems.

## 7. Conclusions

Our contribution to this special issue presents one approach to introducing musical haptics into artistic practices based on immersive multimodal technologies. Our analysis of user experiences highlighted statistically significant differences between haptic and non-haptic scenarios for measures of attractiveness, efficiency, stimulation, and novelty. However, no statistical significance was observed for user experiences of presence and the subcategories of the Presence Questionnaire scales. Embracing musical haptics will assist in connecting artists and audiences, making sharing imaginative content and mutual touch experiences easier. The use of haptic feedback may go beyond the singular, personal, or musical creativity to convey digital media evocative of the past experiences of a musician, an ensemble, or an audience. Musicians will be enabled to communicate performance information, expressing their collective playing experience and creating a shared touch between musicians and audience members. The future development and inclusion of such interfaces in music will rely on musicians' acceptance of these devices and the audience's ability to access them in VR.

**Author Contributions:** G.W.Y.: Conceptualization; methodology; software; validation; formal analysis; investigation; resources; data curation; writing—original draft preparation; writing—review and editing; visualization; supervision; project administration; funding acquisition. N.O.: Conceptualization of VR experience; writing—original draft preparation. M.F.V.: Software (Unity development); Writing—original draft preparation. R.M.D.: Supervision of author three; funding acquisition for author three. A.S.: Conceptualization of VR experience; writing—original draft preparation; V-SENSE project administration; funding acquisition for V-SENSE. All authors have read and agreed to the published version of the manuscript.

**Funding:** This publication has emanated from research conducted with the financial support of the Horizon Europe Framework Program (HORIZON) under grant agreement: 101070109. This work was part supported by the Irish Research Council: [Grant Number IRC-21/PATH-A/9446].

**Data Availability Statement:** The data presented in this study are available on request from the corresponding author.

**Conflicts of Interest:** The authors declare no conflict of interest.

## Notes

1    https://www.newpagans.com/.
2    https://www.volograms.com/.

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
