# Peer review of "Feel the Music!—Audience Experiences of Audio–Tactile Feedback in a Novel Virtual Reality Volumetric Music Video"

_arts, 2023_

Round 1

Reviewer 1 Report

This seems a very relevant paper, on an interesting topic that is not often covered in research on haptic feedback in music - the use of haptic feedback in VR volumetric videos. VR has always had a strong emphasis on the visual modality, supported by audio, and only in a much lesser extend on the tactual modalities. As such, this work is an important contribution. 

The research has carried out with a good level of rigour, in methodology and implementation, in an appropriate mix of quantitative and qualitative research. The resulting data is analysed presented as convincing data, and compelling statements of the subjects. 

Overall it is a good paper, but a few improvements are suggested for the structure, and some details. 

In the Background section (2), the Haptic section is a bit muddled with the different terms as defined in tactual perception literature (but this is often the case in applied haptics work), vibrations perceived through the skin (cutaneous sensitivity) is variously labelled as "haptic", "tactual", and "vibrotactile" (and "proprioceptive" in the diagram but perhaps this refers to another part of the system) - but these are all different submodalities of tactual perception! A bit more clarity and consistency in on this topic would be desirable. 

The methodology section (3) is a bit short, it mainly reports the details of the set-up (and could reveal a bit more, for instance on the "Logitech X100" - here we could do with a bit more of a description of this Bluetooth speaker, and the Glove although this refers to other papers about this custom made device, more details about this design (and how it relates to other musical - haptic gloves that have been around) could be useful). 

The Discussion section (5) is very long, this is appropriate as there is a lot to discuss, but it could be structured in sub-sections. The final sections, on Constraints and Future Work (6) and Conclusions (7) are a bit short, and the Conclusions section also brings in new perspectives and literature which might need a bit more reflection at this point. 

A thorough proof-read is desirable, overall the paper is very well written but there are a few points were it seems rushed and details are missing. 

Many figure captions are incomplete. 

While the references are mostly appropriate and inclusive, the References section is a bit sloppy, many references are incomplete; from small details (such as Brenda Laurel's Computers as Theatre - this is the second edition) to larger omissions (reference 24 only mentions a NIME conference). 

Author Response

Reviewer 1.

This seems a very relevant paper, on an interesting topic that is not often covered in research on haptic feedback in music - the use of haptic feedback in VR volumetric videos. VR has always had a strong emphasis on the visual modality, supported by audio, and only in a much lesser extend on the tactual modalities. As such, this work is an important contribution.

Author response: Thank you!

The research has carried out with a good level of rigour, in methodology and implementation, in an appropriate mix of quantitative and qualitative research. The resulting data is analysed presented as convincing data, and compelling statements of the subjects.

Author response: We agree with the reviewer's assessment.

Overall it is a good paper, but a few improvements are suggested for the structure, and some details.

In the Background section (2), the Haptic section is a bit muddled with the different terms as defined in tactual perception literature (but this is often the case in applied haptics work), vibrations perceived through the skin (cutaneous sensitivity) is variously labelled as "haptic", "tactual", and "vibrotactile" (and "proprioceptive" in the diagram but perhaps this refers to another part of the system) - but these are all different submodalities of tactual perception! A bit more clarity and consistency in on this topic would be desirable.

Author response: The Background subsection — Haptics — has been rewritten to try and clarify our meaning.

The methodology section (3) is a bit short, it mainly reports the details of the set-up (and could reveal a bit more, for instance on the "Logitech X100" - here we could do with a bit more of a description of this Bluetooth speaker, and the Glove although this refers to other papers about this custom made device, more details about this design (and how it relates to other musical - haptic gloves that have been around) could be useful).

Author response: Further details outlining the technical specifications of the audio-tactile Glove and the X100 speaker have now been added to this section. We have also included references to other haptic gloves in the music technology space.

The Discussion section (5) is very long, this is appropriate as there is a lot to discuss, but it could be structured in sub-sections.

Author response: Subsections have been added.

The final sections, on Constraints and Future Work (6) and Conclusions (7) are a bit short, and the Conclusions section also brings in new perspectives and literature which might need a bit more reflection at this point.

Author response: This has not been completed due to time and space constraints. The authors believe that these new perspectives and literature provide further insight into how the work will be built upon in the future.

A thorough proof-read is desirable, overall the paper is very well written but there are a few points were it seems rushed and details are missing.

Author response: More care has been taken to edit and remove mistakes.

Many figure captions are incomplete.

Author response: Care has been taken to provide more information in figure captions

While the references are mostly appropriate and inclusive, the References section is a bit sloppy, many references are incomplete; from small details (such as Brenda Laurel's Computers as Theatre - this is the second edition) to larger omissions (reference 24 only mentions a NIME conference).

Author response: The reference section has been revisited to provide correct information.

Reviewer 2 Report

Overall this paper needs further editing. It is unclear your position on embodied vs disembodied interaction using volumetric video and haptic feedback in your articulation of background. More articulation needed on methodology over VR overview. Overall the sample size of 20 is a small sample. It does not factor age or cultural bias. Overall results data needs to be further synthesized and needs editing for coherence. Conclusion and next steps need further elaboration. Overall major edits needed to present research.

Author Response

Reviewer 2.

Overall this paper needs further editing.

Author response: Agreed. We hope that the revamped and targeted editing satisfies Reviewer 2.

It is unclear your position on embodied vs disembodied interaction using volumetric video and haptic feedback in your articulation of background.

Author response: In the background section, we introduce topics and concepts related to our work. In the discussion section, we unpack these definitions and discuss them further within the context of the presented experiment. We have added to the discussion that further research is required to find orientation around topics of embodied vs. disembodied interaction using volumetric video and haptic feedback.

More articulation needed on methodology over VR overview.

Author response: More information pertaining to the VR experience has been added to the methodology section.

Overall the sample size of 20 is a small sample. It does not factor age or cultural bias.

Author response: The power and experimental participant numbers were calculated using a GPower analysis. This analysis has been added to further support the sample size. We also include reference to HCI materials that justify the number of participants chosen.

Overall results data needs to be further synthesized and needs editing for coherence.

Author response: The results have been edited for coherence.

Conclusion and next steps need further elaboration.

Author response: This section has had minor editorial due to time and space constraints.

Overall major edits needed to present research.

Author response: Further editing has been completed to provide clarity.

Reviewer 3 Report

The paper reports on a study of the influence of haptic feedback (vibrotactile) while experiencing volumetric music videos In VR with the hypotheses that 1) such feedback will enhance the experience and 2) can augment the feelings of presence.

The paper is well written and the general aim of the study is clear, but there are a number of aspects which are not reported in a transparent manner.

It starts with the second hypothesis of the “augmentation of the sense of presence”. I am not sure what the authors refer to here. As presence is measured we might want to know if the sense of presence increased or decreased (qualitative) or what participants commented on how their sense of being part of the virtual scene was impacted (qualitative). Similarly, I suggest to clarify to which aspects “enhancement” is referred to (first hypothesis): Enjoyment? Entertainment? Listening skills? Clearly defined hypotheses are vital for choosing the appropriate methodology and structuring the discussion of the results.

A lack of rigour is also present in the related work where it is not always clear who the tangent user of the system is - the musician or the audience? (e.g. line 133 to 139) Here a clear position would help the reader to understand the authors’ perspective. Another aspect of the related work is the citation of a rather narrow (and connected) circle of authors in regards to VR applications such as educational (Zermann et al.), physiological (Young et al.) and cultural (O’Dwyer et al.). Here I would like to see also some examples from other groups of researchers. Please include appropriate references throughout the Background and Related Work section, e.g. for statements in 230 to 232 or 266 to 287. Also double-check that all references are included correctly in the reference list: e.g. Schrepp et al., 2017 and Laugwitz et al., 2008 are missing, Witmar and Singer is wrongly spelled (172), the reference to the ITQ is missing (369), the provided UQO link does not work … and the formatting of the reference list is inconsistent in places.

More importantly the study needs to be described in more detail such as video content and relation to haptic stimuli, study design (between/within subjects), exposure time, procedure, duration for procedure, form of navigation, ethical approval, …   

In regards to reporting the results I would suggest not to repeat the test statistic in the text which is already listed in the table, but include for each significant item the effect size of Cohen’s d (https://en.wikiversity.org/wiki/Cohen%27s_d) instead of Eta-Squared.

The presentation of the presence questionnaire results lacks a statement of the overall sense of presence. Apart that there were no differences found, did participants feel present or not? Are there any benchmarks for the ITQ? For further studies I would recommend to look into other standardised presence questionnaires which might be more suitable in the given context of a multi-modal experience:

IPQ: Schubert, T., Friedmann, F., & Regenbrecht, H. (2001). The experience of presence: Factor analytic insights. Presence: Teleoperators & Virtual Environments, 10(3), 266-281.

ITC-SOPI: Lessiter, J., Freeman, J., Keogh, E., & Davidoff, J. (2001). A cross-media presence questionnaire: The ITC-Sense of Presence Inventory. Presence: Teleoperators & Virtual Environments, 10(3), 282-297.

Table 3 seemed a bit out of place and did not add to the discussion as the interview findings are discussed in length later on.

A serious issue in the discussion section is the lack of structure and that the authors overstate the findings. One example is: “Hedonistically, both stimulation, and novelty provided an exciting and motivating experience that was fun to use, and the design of the creative experience caught the users' interest.” (440, 441) This is an inappropriate conclusion as neither excitement, fun or users’ interest was measured. I urge the authors to be true to the facts and connect their findings with the existing literature in the discussion section. Subheadings for the different aspects discussed would be helpful for the reader to understand the structure. For example, I found line 427 to 431 quite confusing as there was no motivation provided. Also, the discussion of task performance (439) was not motivated as there were no tasks to be completed in my understanding.

Furthermore, the paper needs to include a thorough Limitations section and convincing conclusions which are informed by the findings of the study. Currently both sections feel quite superficial and left me wondering how the conclusions were supported by the study design and findings and what limitations were present apart from the stated constraints.

All-in-all an interesting study which would profit from a more rigorous reporting style.

Author Response

Reviewer 3.

The paper reports on a study of the influence of haptic feedback (vibrotactile) while experiencing volumetric music videos In VR with the hypotheses that 1) such feedback will enhance the experience and 2) can augment the feelings of presence.

The paper is well written and the general aim of the study is clear, but there are a number of aspects which are not reported in a transparent manner.

It starts with the second hypothesis of the "augmentation of the sense of presence". I am not sure what the authors refer to here. As presence is measured we might want to know if the sense of presence increased or decreased (qualitative) or what participants commented on how their sense of being part of the virtual scene was impacted (qualitative). Similarly, I suggest to clarify to which aspects "enhancement" is referred to (first hypothesis): Enjoyment? Entertainment? Listening skills? Clearly defined hypotheses are vital for choosing the appropriate methodology and structuring the discussion of the results.

Author response: The sense of presence measured refers to the metrics provided by Witmer & Singer (1998). We have changed our hypothesis to reflect this focus and how we report these findings in the discussion and conclusions section.

A lack of rigour is also present in the related work where it is not always clear who the tangent user of the system is - the musician or the audience? (e.g. line 133 to 139) Here a clear position would help the reader to understand the authors' perspective.

Author response: The user of our system is explicitly identified as the audience. We have included references to the performer in the introduction and background section as this research is relevant to using haptics in musical interactions. A new paragraph has been added to further scope the reader into the audience focus of this research.

Another aspect of the related work is the citation of a rather narrow (and connected) circle of authors in regards to VR applications such as educational (Zermann et al.), physiological (Young et al.) and cultural (O'Dwyer et al.). Here I would like to see also some examples from other groups of researchers.

Author response: The research presented here is a cumulation of work from multiple projects carried out by the current research team over many years. We believe it is crucial to demonstrate that this is cumulative research, not nascent. Other research relating to the submitted work is already included alongside our own to provide further context. There are over 80 references to current and past work in the manuscript, the majority being the work of other researchers. 

Please include appropriate references throughout the Background and Related Work section, e.g. for statements in 230 to 232 or 266 to 287.

Author response: Reference to fundamental theories added.

Also double-check that all references are included correctly in the reference list: e.g. Schrepp et al., 2017 and Laugwitz et al., 2008 are missing, Witmar and Singer is wrongly spelled (172), the reference to the ITQ is missing (369), the provided UQO link does not work … and the formatting of the reference list is inconsistent in places.

Author response: The missing references for Schrepp et al. (2017) and Laugwitz et al. (2008) have been added. The spelling of "Witmer" has been corrected. The link to the ITQ has been updated. Further editing has been carried out in the reference section to correct formatting.

More importantly the study needs to be described in more detail such as video content and relation to haptic stimuli, study design (between/within subjects), exposure time, procedure, duration for procedure, form of navigation, ethical approval, …  

Author response: Reference to our institute's research ethics committee and their approval has been added. A detailed description and link to a video of the experience have been added to the methodology. Further information relating to the vibrotactile feedback device has also been included.

In regards to reporting the results I would suggest not to repeat the test statistic in the text which is already listed in the table, but include for each significant item the effect size of Cohen's d (https://en.wikiversity.org/wiki/Cohen%27s_d) instead of Eta-Squared.

Author response: This appears to be more of a stylistic difference in reporting results. We have removed the mean and standard deviation figures from the results writing section, keeping them in table fomat for quick access. Using Eta squared for reporting effect sizes for T-test analyses is also acceptable in many disciplines. We have included Cohen's d as our effect-size measure, as requested.

The presentation of the presence questionnaire results lacks a statement of the overall sense of presence. Apart that there were no differences found, did participants feel present or not? Are there any benchmarks for the ITQ? For further studies I would recommend to look into other standardised presence questionnaires which might be more suitable in the given context of a multi-modal experience:

IPQ: Schubert, T., Friedmann, F., & Regenbrecht, H. (2001). The experience of presence: Factor analytic insights. Presence: Teleoperators & Virtual Environments, 10(3), 266-281.

ITC-SOPI: Lessiter, J., Freeman, J., Keogh, E., & Davidoff, J. (2001). A cross-media presence questionnaire: The ITC-Sense of Presence Inventory. Presence: Teleoperators & Virtual Environments, 10(3), 282-297.

Author response: This has now been addressed in the results section.

Table 3 seemed a bit out of place and did not add to the discussion as the interview findings are discussed in length later on.

Author response: Table 3 provides the reader with a general overview of the qualitative feedback given by the participants. The individual factors that pertain to the study are expanded upon in the discussion to focus on the most relevant areas of debate, making them distinct in how they communicate the findings to the reader.

A serious issue in the discussion section is the lack of structure and that the authors overstate the findings. One example is: "Hedonistically, both stimulation, and novelty provided an exciting and motivating experience that was fun to use, and the design of the creative experience caught the users' interest." (440, 441) This is an inappropriate conclusion as neither excitement, fun or users' interest was measured.  I urge the authors to be true to the facts and connect their findings with the existing literature in the discussion section.

Author response: What the reviewer states are a "serious issue" is incorrect. The individual factors of the UEQ measure these factors distinctly, so it is not inappropriate to conclude these outcomes as they are precisely measured. Further details on the questionnaire can be found on the UEQ website here: https://www.ueq-online.org/.

Subheadings for the different aspects discussed would be helpful for the reader to understand the structure. For example, I found line 427 to 431 quite confusing as there was no motivation provided. Also, the discussion of task performance (439) was not motivated as there were no tasks to be completed in my understanding.

Author response: The discussion has been divided into subsections to give the reader some expectations of what is being discussed. The methodology now contains further information on their "tasks" and how they interacted with the content.

Furthermore, the paper needs to include a thorough Limitations section and convincing conclusions which are informed by the findings of the study. Currently both sections feel quite superficial and left me wondering how the conclusions were supported by the study design and findings and what limitations were present apart from the stated constraints.

Author response: The "Limitations and Future Work" and "Conclusions" section has been edited to provide further clarity on the contributions of this work to the field of haptics in art practice and give the reader a clearer understanding of the results.

All-in-all an interesting study which would profit from a more rigorous reporting style.

Author response: Thank you! We hope that our additional editing and writing have satisfied the reviewers.

Round 2

Reviewer 3 Report

Thank you for the efforts to improve the paper. I enjoyed checking out the video as it gave me a much better sense what the participants where presented with.

I also checked out the UEQ link and I agree that appropriate to conclude on excitement, enjoyability and interest as those are items of the UEQ. Still I feel that the jump from Attractiveness, Efficiency, Stimulation, and Novelty (505) to Excitement, Fun and Interest reads quite inconclusive and would need a bit further explanation. A short note on what basis this conclusion was draw, e.g. "as the UEQ specifically measures enjoyment, ... which are loading on the subscales of ... we can conclude ... " would have helped me to see the connection.

Please check the ITQ results and discussion:

- The mean for the Haptic subscale seem exceptionally low (which seems counterintuitive to me) and does not result in a statistically significant result?

- The t-value for the Total scores (0.87) does not match the value in the table (483).

- Please clarify "the magnitude of the differences in the means was large" (484). How did you arrive on this conclusion and where is Cohens-d coming from? In fact, this sentence should be deleted as there was no statistical difference - so no need to elaborate.

Good work and all the best!

Author Response

Reviewer 3

Thank you for the efforts to improve the paper. I enjoyed checking out the video as it gave me a much better sense what the participants where presented with.

Author Response: We appreciate the reviewer's comments and agree that the suggested additional contributions and editing have improved the paper.

I also checked out the UEQ link and I agree that appropriate to conclude on excitement, enjoyability and interest as those are items of the UEQ. Still I feel that the jump from Attractiveness, Efficiency, Stimulation, and Novelty (505) to Excitement, Fun and Interest reads quite inconclusive and would need a bit further explanation. A short note on what basis this conclusion was draw, e.g. "as the UEQ specifically measures enjoyment, ... which are loading on the subscales of ... we can conclude ... " would have helped me to see the connection.

Author Response: We have included a short note, as suggested.

Please check the ITQ results and discussion:

- The mean for the Haptic subscale seem exceptionally low (which seems counterintuitive to me) and does not result in a statistically significant result? — corrected

- The t-value for the Total scores (0.87) does not match the value in the table (483). — corrected

- Please clarify "the magnitude of the differences in the means was large" (484). How did you arrive on this conclusion and where is Cohens-d coming from? In fact, this sentence should be deleted as there was no statistical difference - so no need to elaborate. — removed

Author Response: These oversights have been corrected or removed.